# Mitochondria in Retinal Ganglion Cells: Unraveling the Metabolic Nexus and Oxidative Stress

**DOI:** 10.3390/ijms25168626

**Published:** 2024-08-07

**Authors:** Tsai-Hsuan Yang, Eugene Yu-Chuan Kang, Pei-Hsuan Lin, Benjamin Ben-Chi Yu, Jason Hung-Hsuan Wang, Vincent Chen, Nan-Kai Wang

**Affiliations:** 1Department of Education, Chang Gung Memorial Hospital, Linkou Medical Center, Taoyuan 33305, Taiwan; yangth@cgmh.org.tw; 2College of Medicine, National Yang Ming Chiao Tung University, Taipei 11217, Taiwan; 3Department of Ophthalmology, Chang Gung Memorial Hospital, Linkou Medical Center, Taoyuan 33305, Taiwan; yckang@cgmh.org.tw; 4College of Medicine, Chang Gung University, Taoyuan 33302, Taiwan; 5Graduate Institute of Clinical Medical Sciences, College of Medicine, Chang Gung University, Taoyuan 33302, Taiwan; 6Department of Ophthalmology, Edward S. Harkness Eye Institute, Columbia University Irving Medical Center, Columbia University, New York, NY 10032, USA; y04507@ms1.ylh.gov.tw (P.-H.L.); jason.wang@gwu.edu (J.H.-H.W.); 22nb76@queensu.ca (V.C.); 7National Taiwan University Hospital, Yunlin 640203, Taiwan; 8Fu Foundation School of Engineering & Applied Science, Columbia University, New York, NY 10027, USA; bby2108@columbia.edu; 9Columbian College of Arts and Sciences, George Washington University, Washington, DC 20052, USA; 10Faculty of Health Sciences, Queen’s University, Kingston, ON K7L 3N9, Canada; 11Department of Ophthalmology, Edward S. Harkness Eye Institute, Vagelos College of Physicians and Surgeons, Columbia University Irving Medical Center, Columbia University, New York, NY 10032, USA

**Keywords:** mitochondria, retinal ganglion cells, oxidative stress, metabolism, glaucoma, autosomal dominant optic atrophy, retinopathy, antioxidants, gene therapy, mitochondrial transplantation

## Abstract

This review explored the role of mitochondria in retinal ganglion cells (RGCs), which are essential for visual processing. Mitochondrial dysfunction is a key factor in the pathogenesis of various vision-related disorders, including glaucoma, hereditary optic neuropathy, and age-related macular degeneration. This review highlighted the critical role of mitochondria in RGCs, which provide metabolic support, regulate cellular health, and respond to cellular stress while also producing reactive oxygen species (ROS) that can damage cellular components. Maintaining mitochondrial function is essential for meeting RGCs’ high metabolic demands and ensuring redox homeostasis, which is crucial for their proper function and visual health. Oxidative stress, exacerbated by factors like elevated intraocular pressure and environmental factors, contributes to diseases such as glaucoma and age-related vision loss by triggering cellular damage pathways. Strategies targeting mitochondrial function or bolstering antioxidant defenses include mitochondrial-based therapies, gene therapies, and mitochondrial transplantation. These advances can offer potential strategies for addressing mitochondrial dysfunction in the retina, with implications that extend beyond ocular diseases.

## 1. Introduction

Retinal ganglion cells (RGCs) are located in the inner retina and play a crucial role in transmitting visual signals from the retina to the brain. These signals facilitate conscious visual perception and non-image-forming processes, such as the vestibulo-ocular reflex, optokinetic reflex, pupillary light reflex, and circadian rhythm regulation. RGCs are of two types, namely intrinsically photosensitive RGCs (ipRGCs), which contain the melanopsin photopigment, and non-ipRGCs [1]. Non-ipRGCs contribute to conscious visual perception by integrating signals from first-order photoreceptors (rods and cones) and second-order bipolar cells. During this process, photoreceptors convert light stimuli into graded potentials through phototransduction, which results in either depolarization of ON bipolar cells or hyperpolarization of OFF bipolar cells [2]. Subsequently, action potentials generated in non-ipRGCs travel along the optic nerve, a collection of RGC axons, from the retina to the lateral geniculate nucleus. This pathway ultimately enables the formation of conscious visual perception in the visual cortices [3]. In addition to conscious visual perception, ipRGCs participate in non-image-forming processes. They relay action potentials, which are generated directly from light stimuli or indirectly through the photoreceptor–bipolar cell–ipRGC pathway to various brain regions [4,5]. These processes involve the pretectal nucleus and the Edinger–Westphal nucleus for the pupillary light reflex; the suprachiasmatic nucleus in the hypothalamus for circadian rhythm regulation; the medial vestibular nucleus for the vestibulo-ocular reflex; and the oculomotor, trochlear, and abducens nuclei for the optokinetic reflex [6,7,8].

RGCs have high metabolic demands that are influenced by the efficiency of mitochondria. Thus, mitochondrial dysfunction can have profound implications for RGC viability and visual function. Mitochondria contain mitochondrial DNA (mtDNA), and mitochondrial function depends on a coordinated interplay between proteins encoded by mtDNA and those encoded by nuclear DNA (nDNA). Mutations or disruptions in either mtDNA or nDNA can lead to mitochondrial dysfunction, which can affect cellular bioenergetics and potentially cause various mitochondrial diseases.

## 2. Metabolic Demands of RGCs

RGCs possess unique characteristics that require increased metabolism to maintain normal visual functions. In addition to glycolysis, RGCs depend on mitochondria to fulfill their metabolic requirements through oxidative phosphorylation (OXPHOS). In addition, metabolic interdependence between RGCs and astrocytes helps with meeting the specific metabolic demands of RGCs.

### 2.1. High Metabolic Demands of RGCs

RGCs are specialized cells with a unique cytoarchitecture, and they function as primary conduits for transmitting visual signals from the retina to the brain [9,10]. They possess distinctive features, such as extensive dendritic arborization, large soma, and a unique axonal structure. The extensive dendrites of RGCs integrate signals from various retinal cells in the inner plexiform layer, whereas the soma initiates action potentials [9]. Figure 1 represents the cytoarchitecture of RGCs that contribute to the high metabolic demands.

RGCs have lengthy axons because they frequently transmit visual information and thus require efficient adenosine triphosphate (ATP) production to meet their high energy demands [11]. Certain segments of RGC axons engage in saltatory conduction, a process that enhances energy efficiency by reducing the frequency of action potentials and clustering sodium channels. RGC axons extend both within and outside of the eyeball. The inner portion of the eyeball requires more mitochondria to meet energy demands because it lacks a myelin sheet, which prevents the occurrence of saltatory conduction [11,12,13,14]. By contrast, the area outside of the eyeball has fewer mitochondria but is more effective in energy production because of myelination, which facilitates saltatory conduction [15]. Thus, RGCs inside the eyeball require more ATP to produce action potentials for transmitting visual information to the brain. Moreover, RGCs release neurotransmitters, such as glutamate, along the visual processing pathway, which further increases their energy requirements [16]. Because of these energy-intensive processes, RGCs are densely packed with mitochondria to ensure continuous ATP production to meet the cell’s demands.

The axons of RGCs are long and thus require more ATP for propagation through Na/K-ATPase and for trafficking vesicles and mitochondria [17]. Moreover, the long axons of RGCs are susceptible to injury, which can lead to increased superoxide production in the soma and neighboring axons and trigger apoptotic signaling in RGCs [18].

The distinct functions and cytoarchitecture of RGCs explain their high metabolic demands. The retina is among the most metabolically active tissues [19,20]. Approximately one-third of the brain’s cortical surface is involved in visual processing, and 90% of all sensory signals integrated within the brain are of visual origin [9]. These facts highlight the critical role of mitochondria in the functioning of RGCs.

### 2.2. Role of Mitochondria in Meeting the Metabolic and Physiological Demands of RGCs

Mitochondria play an essential role in meeting the metabolic demands of RGCs because they ensure optimal ATP production through OXPHOS and mitochondrial dynamics. After glycolysis in cytoplasm, in which two ATP molecules are produced for each glucose molecule oxidized into pyruvate, pyruvate can either be anaerobically converted into lactate in the cytoplasm without ATP production or enter the mitochondria for the tricarboxylic acid (TCA) cycle and OXPHOS [21].

Mitochondrial dynamics, including fusion, fission, transportation, biogenesis, and mitophagy, enable cells to adapt to physiological or stress-induced metabolic changes. Fission involves the division of mitochondria to remove damaged components. By contrast, fusion involves the merging of multiple mitochondria, which enables them to share essential components to supplement deficient ones. The interplay between fission and fusion dynamics determines the morphology of mitochondria [20]. Mitochondrial transportation facilitates the distribution of mitochondria to regions within a cell that have higher energy demands [22]. Mitobiogenesis refers to the generation of new mitochondria from preexisting ones, which increases their number or size. Mitophagy, the process of removing damaged mitochondria, plays a vital role in maintaining mitochondrial quality under normal and stress conditions.

In addition to generating energy, mitochondria protect the integrity of RGCs by regulating calcium levels, which are essential for synaptic transmission, neurotransmitter release, and neuronal function [17,23]. Furthermore, mitochondria maintain redox homeostasis in RGCs by effectively controlling reactive oxygen species (ROS) levels and managing oxidative stress [24]. 

Mitochondria play an indispensable role in meeting the high metabolic demands of RGCs, which align with the retina’s exceptionally high oxygen requirements per unit weight, which are some of the highest in the body [9,19]. The molecular basis of each mitochondrial physiological process and how deviations in these processes affect the functions, morphology, and survival of RGCs are discussed in the following sections.

### 2.3. Metabolic Interdependence between RGCs and Astrocytes

The astrocyte–neuron lactate shuttle (ANLS) hypothesis proposed by Pellerin et al. [25] posits that to meet neuronal metabolic demands, astrocytes provide lactate to neurons, where they serve as an energy substrate. This process is facilitated by astrocytes with low levels of pyruvate dehydrogenase (PDH) activity, which limits the conversion of pyruvate to acetyl-CoA, and high levels of lactate dehydrogenase 5 (LDH5), which promotes the conversion of pyruvate to lactate. This lactate is then transported from astrocytes to neurons, where LDH1 efficiently converts it back to pyruvate, which serves as the substrate for OXPHOS in neurons [26,27]. 

Recent studies demonstrate that the ANLS is not only exhibited in the brain but also in the RGCs. In the retina, aquaporin 9 (AQP9) and monocarboxylate transporters (MCTs) transport lactate alongside glucose transporters (GLUTs) for glucose [28,29,30,31,32,33,34]. Mori et al. conducted an in vivo study indicating that AQP9 functions as an ANLS in conjunction with MCTs to support RGC function and survival under physiological and stress conditions [35]. In *Aqp9*-null mice, there is an increase in retinal GLUT1 expression, particularly in the ganglion cell layer, along with decreased intraretinal L-lactate and increased D-glucose concentrations. This shift towards glucose utilization for energy suggests a compensatory response to inefficient lactate transport under physiological conditions. In contrast, under stress conditions induced by optic nerve damage, *Aqp9*-null mice exhibit significantly reduced RGC density compared to wild-type mice, while the intravitreal injection of an MCT2 inhibitor results in a more pronounced decrease in RGC survival in wild-type mice than in *Aqp9*-null mice. This suggests the potential compensatory role of AQP9 in lactate transport when MCT2 function is impaired. 

Another study observed that in an ocular hypertension model, *Aqp9*-null mice show compensatory upregulation of MCT1 and MCT2 in the ganglion cell layer compared to wild-type mice without significant differences in RGC density loss [36]. These findings demonstrate that AQP9 collaborates with MCTs as an ANLS to sustain RGC function and survival.

In summary, RGCs, which have specific and high metabolic demands, substantially depend on the support of neuronal mitochondria and astrocytes for their energy needs and overall cellular homeostasis [37].

## 3. Mitochondrial Dysfunction in RGCs

RGCs are high-energy consumers that highly depend on the spatiotemporal distribution of functional mitochondria being appropriate, making them susceptible to mitochondrial dysfunction. In this section, we discuss the causes and consequences of mitochondrial dysfunction in RGCs, which can lead to specific degenerative diseases. Specifically, we focus on the molecular basis and present the latest experimental evidence.

### 3.1. Factors Contributing to Mitochondrial Dysfunction in RGCs

Mitochondrial dysfunction in RGCs can result from genetic mutations, environmental stressors, and aging.

Because mitochondria are regulated by both nuclear and mitochondrial genomes, mutations in either genome can lead to mitochondrial dysfunction. This dysfunction contributes to the pathogenesis of hereditary optic neuropathies and may increase susceptibility to acquired RGC degeneration [38]. mtDNA, a double-stranded circular genome within the mitochondrial matrix, is maternally inherited because of the elimination of sperm mitochondria during fertilization [12]. mtDNA encodes proteins involved in the electron transport chain (ETC), ribosomal RNA, and transfer RNA, whereas nDNA encodes most of the mitochondrial proteins and transcription factors involved in energy generation, mitochondrial dynamics, biosynthesis, and mtDNA repair [39]. Thus, mutations in either mtDNA or nDNA can result in mitochondrial dysfunction.

In addition to genetic mutations, environmental stressors, such as pollution, toxins, and UV radiation, contribute to mitochondrial dysfunction through oxidative stress [40]. Mitochondria are a major producer of ROS. Thus, mitochondrial biomolecules, such as mtDNA, proteins, carbohydrates, and lipids, are particularly vulnerable to oxidative damage due to their close proximity to ROS-generating sites within mitochondria [41]. Moreover, increased oxidative stress can induce the release of cytochrome c from mitochondria into the cytosol, a crucial step in the activation of the intrinsic apoptotic pathway that leads to cell death [42]. Additional details regarding environmental factors that contribute to oxidative stress are presented in Section 4.

During aging, genetic and epigenetic mutations accumulate due to environmental stressors or replication errors. Unlike nDNA, mtDNA does not have certain key repair mechanisms, such as mismatch repair and ribonucleotide excision repair [43]. Thus, compared with nDNA, mtDNA is more susceptible to the accumulation of mutations, which can lead to mitochondrial dysfunction. Over time, this increased susceptibility contributes to the development of age-related diseases, including acquired RGC degenerations (e.g., glaucoma) [44].

### 3.2. Susceptibility of RGCs to Mitochondrial Dysfunction

RGCs are particularly susceptible to mitochondrial dysfunction because of their specific anatomy, high energy demands, and unique microenvironment. First, direct exposure of RGCs to light and photoreaction processes can result in an imbalance of ROS homeostasis [45]. Second, RGCs have high metabolic demands because of their specialized functions and distinct cellular structure, which require a continuous supply of ATP to maintain functionality. Moreover, the high energy requirement of RGCs increases their susceptibility to ATP depletion and subsequent mitochondrial dysfunction. Third, the microenvironment of RGCs, which includes interactions with adjacent amacrine cells, oligodendrocytes, and bipolar cells, affects the survival of RGCs and their ability to regenerate axons [11,46,47,48].

Photoreceptors, which are responsible for maintaining the dark current and engaging in energy-dependent light transduction, lead to the outer retina having the highest level of oxygen consumption within the retina, indicating robust mitochondrial activity [3,49]. The oxygen consumption gradually decreases as the inner retina is approached and then increases again within its deeper layers [50,51]. This pattern corresponds with the mitochondrial immunostaining presented in Figure 2. Additionally, because of the dual blood supply of the retina, the inner retina, which receives its supply from the central retinal artery, is more vascularized and richer in oxygen than the outer retina, which receives its supply from the choriocapillaris [52]. Thus, in the avascular outer retina, approximately 80% of glucose supplied from the choroid is metabolized through anaerobic glycolysis, which produces lactate. By contrast, in the vascularized inner retina, only 20% of the glucose from the retinal circulation is converted to lactate [53,54].

RGCs and photoreceptors both have substantial energy requirements in the retina, but RGCs are more susceptible to mitochondrial dysfunction for several reasons. First, while photoreceptors mainly rely on aerobic glycolysis occurring in the cytoplasm, RGC dendrites primarily depend on OXPHOS in mitochondria [55,56]. Second, RGCs exhibit a lower level of mitochondrial superoxide dismutase (SOD), an enzyme that converts superoxide anions into less bioactive ROS, than other cell types do [57]. This lower expression of SOD results in a decreased intrinsic antioxidant capacity of RGCs, potentially leading to optic neuropathy. Finally, photoreceptors exhibit higher resilience to mitochondrial loss than RGCs do because of their high mitochondrial density [58].

Mitochondrial dysfunction results in ROS accumulation, which leads to increased oxidative stress and subsequent damage to mitochondrial structure and function. This vicious cycle renders RGCs highly susceptible to mitochondrial dysfunction and oxidative stress.

### 3.3. RGC Degenerative Diseases Associated with Mitochondrial Dysfunction

#### 3.3.1. Hereditary Optic Neuropathies: Pathophysiology

Hereditary optic neuropathies comprise a group of genetically diverse, monogenic diseases characterized by the degeneration of RGC axons [59]. These conditions result from genetic mutations in either mtDNA or nDNA that encode proteins involved in mitochondrial function. Hereditary optic neuropathies can manifest as either isolated optic neuropathies or syndromic diseases that involve the degeneration of multiple systems, which is often accompanied by symptoms other than those of optic neuropathy [14,59]. Proteins encoded by genes associated with hereditary optic neuropathies play roles in five key aspects of mitochondrial function: fission/fusion, cellular respiration, OXPHOS, mtDNA replication, and lipid metabolism [14], as depicted in Figure 3. In Table 1, hereditary optic neuropathies are categorized on the basis of physiological functions primarily altered by mutated genes.

Fission

The *OPA1*-encoded mitochondrial dynamin-like GTPase, located in the inner mitochondrial membrane (IMM), works in conjunction with outer mitochondrial membrane (OMM) proteins encoded by *mitofusin 2 (MFN2)*, *dynamin 1-like (DNM1L)*, and *mitochondrial elongation factor 1 (MIEF1)* to regulate mitochondrial fusion or fission. In response to stress conditions that disrupt OXPHOS, the activity of the overlapping activity with M-AAA protease (OMA1) in the IMM catalyzes the cleavage of OPA1 [61]. This process results in the production of the short-form OPA1 (S-OPA1), creating an imbalance that favors mitochondrial fusion [61]. During this stress-induced mitochondrial hyperfusion, which is a survival response, OMA1 activity is upregulated by a peptidase encoded by *AFG3-like matrix AAA peptidase, subunit 2 (AFG3L2)* in the IMM instead of being downregulated by paraplegin (encoded by the *SPG7* gene) in the OMM [62,63]. These interactions among genes and proteins play a crucial role in maintaining balance between fusion and fission, thus regulating the quality and quantity of mitochondria in response to cellular stresses.

Mutations in genes associated with mitochondrial fusion, including *OPA1*, *AFG3L2*, and *MIEF1*, lead to autosomal dominant optic atrophies (DOAs), specifically optic atrophy 1 (OPA1), optic atrophy 12 (OPA12), and optic atrophy 14 (OPA14), respectively. By contrast, mutations in genes associated with mitochondrial fission, including *DNM1L* and *MFN2*, result in different conditions. Mutations in *DNM1L* cause autosomal dominant, late-onset optic atrophy 5 (OPA5), whereas mutations in *MFN2* result in Charcot-Marie-Tooth Disease (CMT) type 2, which often involves neuropathy as a potential symptom [64,65].

Mutations in the *outer mitochondrial membrane lipid metabolism regulator (OPA3)* have been linked to mitochondrial fragmentation, yet conflicting evidence persists regarding its localization within the IMM or OMM [66,67]. *OPA3* mutations are responsible for autosomal dominant optic atrophy 3 (OPA3), which often presents with cataracts [68].

The *solute carrier family 25, member 46 (SLC25A46)* gene encodes a protein at the OMM that interacts with OPA1, MFN2, and components of the mitochondrial contact site and cristae organizing system (MICOS) complex, which is substantial in cristae maintenance [69,70]. Mutations in the SLC25A46 gene are linked to defective fission and are responsible for CMT disease type 6B (CMT type 6B), which is an autosomal recessive condition that may include optic atrophy as part of the broader syndromic presentation [71].

Cellular Respiratory Function

Gene mutations that disrupt OXPHOS, the mitochondria-associated endoplasmic reticulum membrane (MAM), or the citric acid cycle functions can impair cellular respiration and contribute to the pathogenesis of specific hereditary optic neuropathies.

Dysfunction in complex I of the ETC causes disruptive OXPHOS, leading to inefficient bioenergetics, increased ROS generation, and cytochrome c-mediated apoptosis. Mutations in genes that affect the subunit composition or function of complex I are responsible for Leber’s hereditary optic neuropathy (LHON), optic atrophy 7 (OPA7), and optic atrophy 10 (OPA10). In 1988, LHON was identified to be caused by mutations in mtDNA that affect complex I function [72]. Exhibiting phenotypic similarities to LHON, the autosomal recessive Leber-like hereditary optic neuropathy is caused by mutations in the *DNA J homolog subfamily C member 30 (DNAJC30)* gene. This gene encodes a chaperone protein essential for the repair of complex I [73]. OPA7, which is also autosomal recessively inherited, is caused by mutations in the *transmembrane protein 126A (TMEM126A)* gene [74]. This gene encodes a protein embedded in the IMM crucial for the assembly of complex I [75,76]. *RTN4IP1* encodes proteins localized to the OMM, which comprises an oxidoreductase domain that affects complex I [77]. Mutations in the *RTN4IP1* lead to OPA10, which is an autosomal recessive inherited condition that often presents with additional neurological symptoms, such as epilepsy, chorea, or encephalopathy [78].

MAM is a specialized region that facilitates interaction between mitochondria and the endoplasmic reticulum (ER), playing a crucial role in calcium homeostasis between these organelles [79]. This interaction is vital for regulating ER stress, mitochondrial metabolism, and ATP production [80,81]. The *wolframin ER transmembrane glycoprotein (WFS1)* gene encodes wolframin, a protein embedded in the ER membrane, and comprises MAM [82,83]. Wolframin is ubiquitously expressed but particularly prominent in the optic nerve [84]. Wolfram syndrome type 1, also termed DIDMOAD, is an acronym reflecting a combination of symptoms—diabetes insipidus, diabetes mellitus, optic atrophy, and deafness [85]. This autosomal recessively inherited condition is the second leading cause of hereditary optic atrophy.

In addition to gene mutations that affect OXPHOS and MAM, mutations that disrupt the citric acid cycle contribute to impaired cellular respiration, leading to hereditary optic neuropathies. Optic atrophy 9 (OPA9) is caused by mutations in the *aconitase 2 (ACO2)* gene, which encodes an enzyme in the mitochondrial matrix. This enzyme catalyzes the conversion of citrate to isocitrate in the citric acid cycle. In cases of OPA9, individuals with recessive mutations typically experience more severe vision loss, whereas those with dominant cases often exhibit extraocular symptoms. Mutations in the *ACO2* are the third most common cause of autosomal OA, following those in *OPA1* and *WFS1* [14,86].

Mitochondrial DNA Replication

The *single-stranded DNA-binding protein 1 (SSBP1)* gene is essential for proper mtDNA replication. This gene encodes a component of the mtDNA replisome that binds to and stabilizes the single-stranded displacement loop during mtDNA synthesis [87,88]. Mutations in the SSBP1 gene cause autosomal dominant optic atrophy type 13 (OPA13), which is discussed further in the following section.

Lipid Metabolism

Mutations in lipid metabolism-related genes, including *malonyl CoA:ACP acyltransferase (MCAT)* and *mitochondrial trans-2-enoyl-CoA reductase (MECR)*, are responsible for autosomal recessive optic atrophy 16 (OPA15) and optic atrophy 15 (OPA16), respectively [88,89,90]. 

*MCAT* encodes the enzyme in the mitochondrial matrix that transfers malonate from malonyl-CoA to the acyl carrier protein, mediating the initial step of mitochondrial fatty acid synthesis (mtFAS) [91]. Conversely, the MECR gene encodes the final step of mtFAS, with its end product serving as the substrate for lipoic acid synthesis [92,93].

#### 3.3.2. Hereditary Optic Neuropathy: LHON

LHON is a maternally inherited disease caused by mutations in mtDNA that encodes subunits of complex I of the ETC, also termed NADH dehydrogenase [72,94]. Approximately 90% of LHON cases result from three specific point mutations, namely *MT-ND4 m.11778G>A*, *MT-ND1 m.3460G>A*, and *MT-ND6 m.14484T>C* (listed in order of decreasing frequency and severity) [11]. These mutations compromise complex I function, leading to respiratory impairment and ROS accumulation, which in turn cause degeneration of RGCs in patients with LHON. The typical presentation of LHON involves bilateral, painless central scotomas that develop in male patients aged 15 to 35 years, followed by rapid progression to optic nerve atrophy [59].

LHON is characterized by a male predominance, incomplete penetrance among mutation carriers, and a later onset compared with that of other inherited optic neuropathies [59]. These characteristics suggest the presence of spontaneous compensatory mechanisms for complex I dysfunction. Experimental studies have elucidated the mechanisms underlying these features. For example, estrogen was discovered to activate mitochondrial biogenesis in a hybrid cell model, which potentially explains the higher prevalence of LHON in men [95]. Although most LHON cases are homoplasmic, meaning that all copies of mtDNA harbor mutations [96], environmental factors [97,98] or genetic modifiers [99,100,101] may act as a second hit, which may explain why some carriers remain unaffected for life. In addition, certain mitochondrial haplogroups may predispose individuals to environmental factors associated with LHON, including exposure to rotenone or 2,5-hexanedione [102,103,104,105].

#### 3.3.3. Hereditary Optic Neuropathy: OPA1

*OPA1* is the most common gene responsible for autosomal OA [59]. The majority of *OPA1* mutations cause autosomal DOA, which typically begins in the first or second decade of life as a bilateral, slowly progressive loss of vision [106]. Approximately 20% of DOA cases present with a syndromic phenotype known as DOA-plus, which may lead to the development of additional symptoms, such as deafness, peripheral neuropathy, myopathy with chronic progressive external ophthalmoplegia, ataxia, Parkinsonism, and dementia [59,107]. In addition, rare compound heterozygous mutations in *OPA1* can lead to Behr syndrome [108].

#### 3.3.4. Hereditary Optic Neuropathy: OPA13

Whether autosomal dominant inheritance of OPA13 is due to a dominant-negative effect or haploinsufficiency remains unclear because animal model studies have provided evidence supporting both hypotheses. Jurkute et al. demonstrated that mutations in the *SSBP1* gene disrupt the function of the wild-type protein, suggesting a dominant-negative mechanism for SSBP1 [109]. This observation is consistent with the behavior of SSBP1, a multimeric enzyme in which a defective subunit disrupts the function of the entire complex, indicating a dominant-negative effect [110]. However, Dotto et al. proposed that functional null mutations in *SSBP1* lead to the disease phenotype, supporting the concept of haploinsufficiency in the pathogenesis of OPA13 [111].

Section 3.2 discusses the higher vulnerability of RGCs than photoreceptors in hereditary optic neuropathies where mitochondrial dysfunction is the primary cause. However, in the case of OPA13, both retinal imaging and electrophysiological studies have demonstrated that degeneration occurs in both photoreceptors and RGCs, distinguishing OPA13 from other hereditary optic neuropathies [58,109,111,112]. This may be attributable to the role of the SSBP1 protein in mtDNA repair. Mutations in the *SSBP1* gene may disrupt the protein’s multimeric structure, potentially hindering the repair of mtDNA, which is more prone to mutations than nDNA is. Furthermore, nDNA benefits from robust ribonucleotide excision repair and mismatch repair mechanisms. Thus, the compromise in mitochondrial function caused by mutations in the *SSBP1* gene extends beyond RGCs, affecting photoreceptors and various other cell types [58].

#### 3.3.5. Glaucoma

Glaucoma, the leading cause of irreversible blindness worldwide, is characterized by progressive dysfunction and loss of RGCs [113]. Several risk factors, such as age, elevated intraocular pressure (IOP), and genetic predisposition, contribute to the development of glaucoma by causing bioenergetic insufficiency, impairing blood flow autoregulation, and increasing ocular or systemic oxidative stress [114,115]. Increased IOP has been demonstrated to disrupt the autoregulation of blood flow in the optic nerve, leading to increased intraocular oxidative stress in both human glaucoma cases and animal models [114,116,117].

Mitochondrial dysfunction, characterized by bioenergetic insufficiency, mtDNA damage, defective mitochondrial quality control, and oxidative stress, plays a role in the pathophysiology of glaucoma. Age-related bioenergetic insufficiency increases the susceptibility of RGCs to IOP during the pathogenesis of glaucoma [37]. In animal and in vitro models of increased IOP, prior to detectable optic nerve degeneration, mTOR activation and the subsequent reduction in retinal glycolysis have been demonstrated to result in a high glucose level and a low pyruvate level [37]. This period between the initial detection of elevated IOP and the onset of glaucomatous neurodegeneration offers a therapeutic window for oral supplementation of pyruvate or the use of mTOR-inhibiting rapamycin [24].

There is compelling evidence implicating mitochondrial dysfunction, specifically complex I dysfunction and *OPA1* gene involvement, in glaucoma pathogenesis. Mitochondrial complex I dysfunction in primary open-angle glaucoma (POAG) patients manifests both systemically and locally. Lymphoblasts from these patients exhibit slower growth under mitochondrial stress, reduced complex I activity, and ATP synthesis in vitro [118]. Trabecular meshwork cells display elevated ROS levels, decreased ATP levels, and increased susceptibility to complex I inhibition, promoting apoptosis [119]. Moreover, increased flavoprotein fluorescence observed in eyes with ocular hypertension and POAG suggests early mitochondrial stress in vivo before the clinical onset of structural changes [120].

Beyond mitochondrial complex I dysfunction, the *OPA1* gene’s role in POAG is gaining increasing recognition. Variants and reduced expression of *OPA1*, involved in mitochondrial fission and known for its mutation in DOA, have been linked to glaucoma. Experimental glaucoma models demonstrate that *OPA1* overexpression protects against RGC loss by enhancing mitochondrial fusion and parkin-mediated mitophagy [121]. Conversely, decreased *OPA1* expression is observed in POAG patients. Furthermore, single-nucleotide polymorphisms in the *OPA1* gene, including rs166850 C/T and rs10451941 T/C, are associated with POAG susceptibility, particularly in normal tension glaucoma (NTG) and among Caucasian populations [122].

#### 3.3.6. Divergent Patterns of Vision Loss in LHON and Glaucoma

While mitochondrial complex I dysfunction plays a critical role in both LHON and glaucoma, the differential impact on central versus peripheral vision can be attributed to the metabolic demands and structural differences of RGCs, as well as the specific stressors affecting these cells. Lymphoblasts from POAG patients show milder complex I dysfunction compared to those from LHON patients, exhibiting a slower growth rate under mitochondrial stress conditions and a less pronounced reduction in complex I enzyme activity and ATP synthesis [123]. The heightened dependence on mitochondrial function in LHON compared to PAOG contributes to the differing patterns of vision loss in the two diseases.

In LHON, central vision loss is primarily attributed to the vulnerability of small, non-myelinated axons in the papillomacular bundle, which have high metabolic demands and rely heavily on mitochondrial ATP production [124]. This high dependence on mitochondrial function makes these axons particularly susceptible to oxidative stress and energy deficits resulting from complex I dysfunction. In contrast, glaucoma predominantly affects peripheral vision, with larger, myelinated axons initially more resilient to mitochondrial dysfunction but still susceptible to elevated IOP and oxidative stress [125]. Elevated IOP disrupts blood flow and induces mechanical stress on the optic nerve head, leading to ischemia and oxidative damage that predominantly impacts peripheral RGCs [126].

## 4. Oxidative Stress in RGC Health

### 4.1. Definition of Oxidative Stress

Since the introduction of the concept of oxidative stress in 1985, the understanding and the associated terminology of the concept have evolved, indicating diverse effects of ROS on biological systems in both physiological and pathological contexts [127]. Initially viewed as purely damaging agents, ROS are now recognized as pleiotropic signaling molecules involved in various cellular processes. Furthermore, the definition of oxidative stress has expanded. Oxidative stress is no longer merely considered an imbalance between oxidants and antioxidants within a biological system but is considered to occur as a spectrum. This spectrum ranges from “eustress”, which refers to controlled physiological deviations from the normal steady-state redox balance, to “distress”, in which an excessive level of oxidants causes molecular damage [40,128].

The level of oxidative stress is determined by the degree of deviation from a balanced redox state and is affected by the capacities of both oxidants and antioxidants. ROS are molecular oxygen derivatives that exhibit higher chemical reactivity than molecular oxygen itself does [129]. Low levels of ROS act as signaling molecules that regulate various processes, such as thermogenesis, immunity, cellular proliferation, development, and apoptosis (programmed cell death). By contrast, supraphysiological levels of ROS can damage cellular components, such as lipids, proteins, nucleic acids, and carbohydrates, leading to necrosis (uncontrolled cell degradation). Antioxidants are substances that can delay, prevent, or remove oxidative damage from specific target molecules [129]. The interplay between ROS and antioxidants is critical for maintaining redox balance, ultimately influencing both physiological and pathological cellular outcomes.

### 4.2. ROS Metabolism and Antioxidant Defense Pathway in RGC Degeneration

Figure 4 illustrates the ROS metabolism and antioxidant defense pathway, highlighting the pathways of the oxidative stress biomarkers used in studies on RGC degeneration. The molecules involved in these processes are specified as ROS or antioxidants. Changes in their function or levels can serve as oxidative stress biomarkers in research. 

The figure is organized into four main components: the conversion processes of various ROS, which involve specific enzymes or small molecules; the cofactors responsible for replenishing redox enzymes; the sites where antioxidants function; and the biomarkers used to identify oxidative stress. These four aspects are detailed in Section 4.2.1, Section 4.2.2, Section 4.2.3 and Section 4.2.4.

#### 4.2.1. Conversion of Various ROS

The first aspect depicted in Figure 4 is the conversion of various ROS. These ROS mainly include superoxide anion radicals (O_2_^•−^), hydrogen peroxide (H_2_O_2_), hydroxyl radicals (HO^•^), and hypochlorous acid (HOCl) and secondary ROS, such as organic hydroperoxide (ROOH), malondialdehyde (MDA), 4-hydroxynonenal (4-HNE) generated through the peroxidation of polyunsaturated fatty acids (PUFAs), and peroxynitrite (ONOO^−^) formed from O_2_^•−^ and nitric oxide (NO^•^). Initially, O_2_^•−^, which is an ROS and a precursor to various other ROS, is generated from molecular oxygen (O_2_) through mitochondrial ETC complexes I and III during OXPHOS or through various oxidases, such as NADPH oxidase (NOX) enzymes in peroxisomes [130]. Next, H_2_O_2_ either arises from O_2_^•−^ through SOD or from O_2_ through cytochrome P450 in the ER or by peroxisomal NOX [131]. This forms a “redox triangle” involving mitochondria, ER, and peroxisomes [132]. Subsequently, H_2_O_2_ can either be enzymatically reduced into water (H_2_O), generate HO• through the Fenton reaction in the presence of free iron cations, or produce HOCl through myeloperoxidase in phagolysosomal pathogen killing [133,134].

#### 4.2.2. Cofactors Responsible for Replenishing Redox Enzymes

Figure 4 also highlights cofactors involved in replenishing redox enzymes. Peroxides, such as H_2_O_2_ and ROOH, are secondary messengers for various cellular processes [40,135,136,137]. Physiological peroxide levels are maintained by redox relay mechanisms, which involve the transmission of oxidative signals from one molecule to another through reversible redox reactions, ultimately leading to oxidation of the intended target [40]. Glutathione, thioredoxin, and NADPH transition between their reduced and oxidized states, ensuring that glutathione peroxidase and peroxiredoxin effectively reduce peroxides and thereby regulate their levels [138,139]. Specifically, glutathione peroxidase oxidizes glutathione, whereas peroxiredoxin oxidizes thioredoxin. NADPH, generated in the pentose phosphate pathway through glucose 6-phosphate dehydrogenase, is the primary electron donor that replenishes reduced glutathione and reduced thioredoxin via glutathione reductase and thioredoxin reductase, respectively [140,141].

#### 4.2.3. Components and Mechanisms Involved in Antioxidant Defense Pathways

Another aspect depicted in Figure 4 includes pathways through which antioxidants, including enzymatic and nonenzymatic low-molecular-mass compounds (LMMCs), scavenge ROS [40]. Enzymatic antioxidants, such as SOD, catalase, glutathione peroxidase, peroxiredoxin, and thioredoxin reductase, catalyze ROS removal. By contrast, nonenzymatic LMMC antioxidants, including glutathione, thioredoxin, N-acetylcysteine (NAC), CoQ, vitamins E and C, and phytochemicals, directly neutralize ROS. HO^•^ is the most reactive ROS, surpassing O_2_^•−^ and the less reactive H_2_O_2_ [142]. Therefore, SOD, catalase, glutathione peroxidase, peroxiredoxin, thioredoxin, glutathione, and NAC (a glutathione precursor) are categorized as antioxidants. In addition, metal ion-binding proteins, such as metallothioneins, ferritin, transferrin, lactoferrin, ceruloplasmin, and albumin, function as antioxidants by sequestering free iron ions, reducing their contribution to the Fenton reaction [143]. CoQ acts as an antioxidant by facilitating electron transport from mitochondrial ETC complexes I and II to complex III, thereby reducing O_2_^•−^ generation [144]. Furthermore, vitamin E (α-tocopherol), regenerated by vitamin C (ascorbic acid), scavenges organic peroxyl radicals (ROO^.^) formed during lipid peroxidation. Lipid peroxidation can occur either enzymatically, catalyzed by cyclooxygenases, cytochrome p450s, or lipoxygenases, or through a nonenzymatic chain reaction initiated by HO^•^ or O_2_^•−^ to oxidize PUFAs in cell membranes into ROO^.^, leading to the formation of ROOH and end products, such as MDA or 4-HNE. Phytochemicals, such as flavonoids (e.g., quercetin and catechins), carotenoids (e.g., beta-carotene and lutein), resveratrol, curcumin, allicin, isoflavones (e.g., genistein and daidzein), catechins, polyphenolic acids (e.g., ferulic acid and ellagic acid), and gingerol, donate electrons to directly neutralize ROS [40,145].

#### 4.2.4. Biomarkers for Oxidative Stress

The final aspect, illustrated in Figure 4, pertains to oxidative stress biomarkers, including direct detection of specific ROS and antioxidants or measurement of cellular response to ROS. The focus is on the oxidative stress biomarkers employed in studies examining RGC degeneration.

Certain ROS can be detected using fluorescence-based probes, which emit fluorescence when oxidized by specific ROS, which can provide insights into the spatiotemporal distribution of these short-lived molecules. Examples of such probes include 2–7 dichlorofluorescein diacetate for identifying H_2_O_2_ and dihydroethidium for identifying O_2_^•−^ [146,147].

Increased levels of free iron or upregulation of iron-binding proteins, such as transferrin, ceruloplasmin, and ferritin, serve as markers of oxidative stress [148]. This is because free iron catalyzes conversion of H_2_O_2_ to HO^•^, which is the most reactive among the ROS. In addition, iron triggers ferroptosis by initiating oxidation of PUFAs in a process dependent on it [149].

In addition to direct measurement of specific ROS or antioxidants, the patterns of oxidant-induced modifications of cellular biomolecules can be assessed and can serve as surrogate markers for oxidative stress. Various products of biomolecule oxidation have been used for investigation of oxidative stress in ocular diseases; these products include MDA or 4-HNE from the peroxidation of PUFAs or phospholipids; 7-ketocholesterol, 7β-hydroxycholesterol, 5α,6α-epoxycholesterol, and 5β,6β-epoxycholesterol from peroxidation of cholesterols; 8-hydroxy-2′-deoxyguanosine (8-OHdG or 8-oxodG) from DNA oxidation; 8-hydroxyguanosine from RNA oxidation; the GSH/GSSH ratio; the cysteine/cystine ratio; the protein thiol/disulfide ratio; protein carbonyls (oxidized by HO^•^); 3-nitrotyrosine (oxidized by NO_2_ or peroxynitrous acid); 3-chlorotyrosine (oxidized by HOCl) from oxidized protein or amino acids; and advanced glycation end products from protein glycation [40,150,151].

Specific transcription factors, such as nuclear factor erythroid 2-related factor 2 (Nrf2), that are unregulated by ROS can be detected through immunohistochemistry on retinal sections and can serve as biomarkers for oxidative stress. Translocation of Nrf2 from the cytoplasm to the nucleus, which subsequently leads to upregulation of SOD and catalase expression, indicates increased oxidative levels [152]. This translocation process and subsequent adaptive reactions to oxidative stress, known as hormesis, can be detected through immunohistochemistry on retinal sections, which can reveal novel gene expression profiles with increased antioxidant levels [153].

### 4.3. Impact of Oxidative Stress on RGC Degenerative Diseases

Supraphysiological levels of ROS can result either from the endogenous sources discussed in Section 4.2. or from exogenous environmental factors. Such environmental factors include ionizing radiation, ultraviolet light, sound waves, heat, electromagnetic fields, and exposure to airborne particulate matter [40]. These environmental factors induce oxidative stress through nonenzymatic chemical reactions or upregulation of ROS production pathways while concurrently impairing antioxidant mechanisms. For example, exposure to airborne particulate matter, such as ultrafine atmospheric particulate matter, O_3_, sulfur (SO_x_), and nitrogen oxides (NO_x_), can lead to persistent immune system activation, resulting in the elimination of foreign substances and generation of ROS. This response is triggered by activating the aryl hydrocarbon receptor, which subsequently leads to the expression of cytochrome P450 enzymes [154].

Research findings, including causal and correlational evidence summarized in Table 2, highlight oxidative stress as a significant factor in RGC degenerative diseases.

## 5. Mitochondrial-Based Therapeutic Approaches

With numerous mitochondria working to supply energy to RGCs, mitochondrial dysfunction is inevitable and leads to RGC degeneration. Over time, this degeneration of RGCs leads to various retinal disorders, such as glaucoma and optic neuropathies [11,172]. Although the exact mechanisms underlying the susceptibility of RGCs to mitochondrial dysfunction are not yet fully understood, maintaining the health of mitochondria is critical for preserving vision. Ongoing research is exploring various methods and therapeutic targets that can enable prevention of mitochondrial dysfunction, improve RGC health, and treat diseases associated with RGC degeneration. This section discusses both current and emerging strategies for targeting mitochondrial dysfunction in RGC degeneration.

### 5.1. Antioxidative Therapeutic Target in Mitochondrial Dysfunction

Antioxidants are an emerging therapy for protecting RGCs from oxidative damage. RGCs are highly susceptible to oxidative stress because of their high metabolic activity. Therefore, antioxidants play a pivotal role in neutralizing ROS and preventing oxidative damage by either scavenging ROS or indirectly enhancing the body’s natural defense system [173,174]. In addition, antioxidants can repair oxidative damage, facilitate the regeneration of other antioxidants, and modulate inflammatory responses triggered by oxidative stress [175,176]. Table 3 summarizes the work of previous studies that have investigated the effects of antioxidants on mitochondrial dysfunction.

### 5.2. Potential Therapeutics Involving Mitophagy

Another therapeutic option currently under investigation for mitochondrial dysfunction is mitophagy. Mitophagy refers to a type of autophagy wherein irreparable and damaged mitochondria are selectively degraded by lysosomes [20,192]. Mitochondrial fusion and fission play key roles in mitophagy in that they alter mitochondrial pathways [193]. Dysfunctional mitochondria are prone to producing uncontrollable and excessive levels of ROS, resulting in oxidative stress and cellular damage. Mitophagy acts as a crucial mechanism for mitigating such harm and maintaining mitochondrial quality.

Two forms of mitophagy have been identified: promotion and inhibition. Maintaining a balance between these two processes is crucial for proper cellular function because excessive mitophagy can lead to cell death. Promotion of mitophagy facilitates removal of damaged mitochondria, preventing ROS accumulation and contributing to overall cellular health [194]. By contrast, inhibition of mitophagy reduces elimination of mitochondria, protecting the cell from excessive self-damage. However, additional research is required to fully understand the implications of inhibiting mitophagy, particularly because it may adversely affect vision [195]. The essential role of mitophagy in cellular maintenance and as a defense against mitochondrial dysfunction-related disorders is a focus of current research [196]. Researchers are continuing to explore mitophagy as a potential therapeutic target for mitochondrial disorders and related diseases. Table 4 and Table 5 present the details of studies investigating promotion and inhibition of mitophagy in mitochondrial dysfunction.

### 5.3. Potential Therapeutics Involving Mitobiogenesis

Mitobiogenesis, or mitochondrial biogenesis, is the process through which new mitochondria are created within a cell [211,212], and it is typically activated by cellular stress in response to environmental stimuli. This process involves a highly coordinated effort between nDNA and mtDNA to replicate the specific genome and structure necessary for new mitochondria [213]. In mitobiogenesis, highly coordinated fission and fusion processes divide existing mitochondria and merge them, facilitating production of healthy mitochondria within cells. The transcriptional coactivator PGC1⍺ initiates transcriptional activity, leading to increased replication of mtDNA, production of more mitochondrial proteins, and an overall increase in mitochondrial density [214]. A variety of compounds can stimulate mitobiogenesis. Activation or inhibition of these compounds can enhance mitochondrial density while reducing ROS levels and cellular death. Table 6 summarizes the results of previous studies investigating promotion of mitobiogenesis and its role in mitigating mitochondrial dysfunction.

### 5.4. Potential Therapeutics Involving Mitochondrial Transplantation

Mitochondrial transplantation, also termed MitoPlant, involves transplanting active and healthy mitochondria into specific cells with mitochondrial dysfunction [228,229]. This technique involves direct replacement of dysfunctional mitochondria, enabling rapid elimination of cellular problems and treatment of the underlying disease. MitoPlant can be used to treat both chronic and genetic diseases associated with RGC degeneration and has shown potential in protecting RGCs from cellular death during ocular ischemia [230]. Studies have demonstrated that introducing exogenous mitochondria into retinal ganglion precursor–like cells, particularly under oxidative stress conditions, could enhance mitochondrial uptake and prolong survival by more than three-fold [231]. Table 7 presents the details of previous studies on MitoPlant.

### 5.5. Clinical Trials of Pharmacological and Genetic Therapeutics in LHON Patients

LHON stems from an mtDNA mutation that affects essential components of complex I [94]. The clinical trials for idebenone and gene therapy in LHON patients provide substantial evidence that mitochondrial function is critical for RGC health by demonstrating the impact of mitochondrial-targeted treatments on visual outcomes.

Idebenone, a synthetic analog of coenzyme Q, bypasses dysfunctional complex I and acts as a potent antioxidant [237]. The medication was approved by the European Medicines Agency in 2015 for treating LHON in adults and children, with a recommended dosage of 900 mg per day for patients with a disease duration of up to five years [238].

SS-31 (elamipretide) is another antioxidant shown to restore redox homeostasis, improve mitochondrial quality, and increase exercise tolerance in mice models [184]. Clinical trials evaluating topical 1% elamipretide ophthalmic solutions in LHON patients did not demonstrate significant improvements in BCVA. However, post hoc analyses of the Humphrey automated visual field central region indicate encouraging trends, suggesting a need for further investigation [239].

Lenadogene nolparvovec, administered via intravitreal injection, uses a recombinant adeno-associated virus vector (rAAV2/2-ND4) carrying the wild-type ND4 subunit of complex I to target mitochondria [240]. This gene therapy aims to permanently correct the mitochondrial DNA mutation in LHON patients with the m.11778G>A mutation, which is the most prevalent and causes a severe form of the disease [241].

The clinical trials for idebenone and gene therapy in LHON patients show promising results with tolerable adverse effects, providing evidence for the critical role of mitochondrial function in maintaining RGC health and insights into the translatability of these technologies. However, the variability in patient response and the modest efficacy of idebenone highlight the need for continued research and the development of more effective therapies. Table 8 presents the details of the clinical trials on pharmacological and genetic therapeutics in LHON patients.

### 5.6. Efficacies, Limitations, and Future Perspectives in Mitochondrial-Based and Gene Therapies

This section compares and contrasts the efficacy, limitations, and future perspectives of mitochondrial-based therapeutics, including antioxidants, mitophagy modulation, mitobiogenesis promotion, mitochondrial transplantation, and gene therapies.

The efficacies of different approaches to mitochondrial-based therapies vary. Antioxidants, specifically idebenone and elamipretide, have been translated from animal models to clinical trials that show positive results, demonstrating potential in reducing oxidative stress and stabilizing vision in LHON patients when administered early [184,188,239,242,243]. Strategies promoting mitophagy with agents including rapamycin, metformin, and small molecule S3 enhance RGC survival by removing damaged mitochondria, while mitobiogenesis promotion using compounds such as rosiglitazone, 17β-estradiol, or nicotinamide riboside and improving mitochondrial function have been found in animal models in vivo or human cells in vitro [95,202,217]. Mitochondrial transplantation shows promise in restoring function and survival of RGCs in vivo. Gene therapies utilizing AAV vectors, such as lenadogene nolparvovec, significantly improve visual acuity in LHON patients in clinical trials [238,240,241,245,246].

Despite their potential, these therapeutic strategies face several limitations. The efficacy of antioxidants like idebenone is modest and primarily benefits patients when administered early [244]. The precise regulation of mitophagy is challenging; excessive mitophagy can lead to cell death, while inadequate mitophagy accumulates dysfunctional mitochondria [194,195,196]. Mitochondrial transplantation is still in the experimental stages, with challenges in ensuring transplanted mitochondria’s long-term viability and integration. Gene therapies using AAV vectors carry risks such as intraocular inflammation and potential immune responses, and their long-term safety and efficacy are still under investigation. Additionally, targeting these therapies specifically to RGCs without affecting other retinal cells remains challenging. 

Future research should focus on refining these therapies. Combining antioxidants with other agents, optimizing delivery methods, and balancing mitophagy and mitobiogenesis are promising avenues. Developing biomarkers to monitor mitophagy activity in patients will enable personalized treatment adjustments. Developing non-invasive or minimally invasive delivery methods will enhance patient safety and acceptance. Refining viral vectors to enhance specificity and reduce immune responses is essential for gene therapies, along with exploring non-viral gene delivery systems. Long-term follow-up studies are needed to monitor the durability of therapeutic effects and identify any delayed adverse events.

## 6. Conclusions

The unique anatomy, high energy demand, and specialized microenvironment of RGCs render them susceptible to mitochondrial dysfunction and oxidative stress, which play central roles in the pathogenesis of degenerative diseases affecting RGCs, including glaucoma and hereditary optic neuropathy. Genetic mutations or environmental factors that impair mitochondrial function can lead to an increased imbalance between ROS production and antioxidant defenses, driving RGCs toward oxidative damage and apoptosis. This understanding has prompted the development of therapeutic strategies for enhancing mitochondrial function or mitigating oxidative stress. These strategies include the use of antioxidants, mitochondrial gene therapies, and innovative approaches, such as mitochondrial transplantation, all of which are designed to preserve or restore RGC health and combat degenerative diseases.

Although experimental studies have provided evidence for each aspect discussed in the present study, concerns persist regarding whether classification of RGCs and their associated degenerative diseases has been oversimplified. Such oversimplification can lead to researchers neglecting crucial physiological and functional variations among RGCs, which can obscure distinct pathological mechanisms relevant to different RGC subtypes [248] and thus impede the development of targeted therapeutic strategies tailored to specific RGC subtypes. In addition, the current broad categorization of diseases such as glaucoma and hereditary optic neuropathies fails to account for their inherent heterogeneity; consequently, it may lead to conclusions that lack universal applicability across all disease phenotypes and genetic bases [55]. These observations underscore the importance of the perspectives that are adopted to understand the pathophysiology and therapeutic approaches for RGC degenerative diseases associated with mitochondrial dysfunction or oxidative stress. A more detailed analysis of the specific needs and responses of different RGC subtypes and disease categories could substantially enhance the understanding of this topic and lead to more tailored and effective therapeutic approaches with improved specificity and success rates.

Future studies should address the challenge of translating insights from in vitro studies to in vivo settings. Although in vitro experiments are valuable, they often fail to capture living organisms’ intricate interactions and environmental complexities, potentially resulting in overestimation of therapeutic efficacy when findings are transitioned to clinical applications. Thus, future studies should focus on refining experimental models to ensure that they better mimic physiological conditions and thus enhance the relevance and predictive power of in vitro findings. In addition, the use of animal models in research requires careful consideration because of inherent physiological and genetic differences between species. Although these models can provide valuable insights, they have limitations with respect to replicating human disease manifestations and treatment responses. Recognizing these limitations is essential, and advancements in technology, such as the development of more human-relevant animal models, organoids, and patient-derived cell cultures, can help bridge the translational gap and enhance the applicability of preclinical research findings to human conditions [11,249].

## Figures and Tables

**Figure 1 ijms-25-08626-f001:**
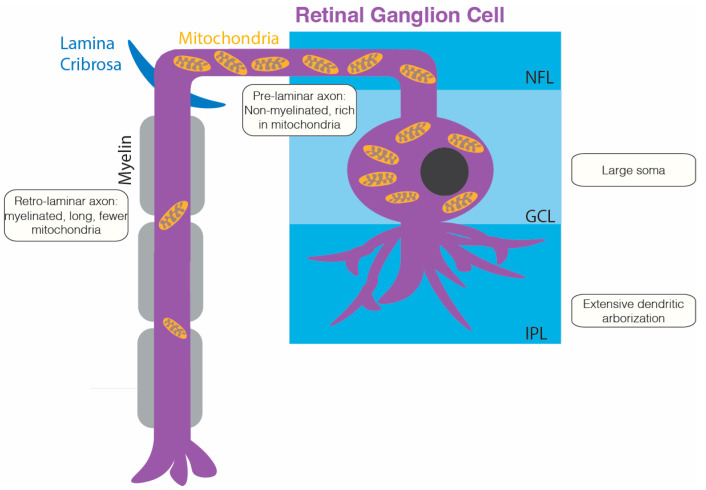
The Cytoarchitecture of Retinal Ganglion Cells Contributing to High Metabolic Demands. IPL, inner plexiform layer; GCL, ganglion cell layer; NFL, nerve fiber layer.

**Figure 2 ijms-25-08626-f002:**
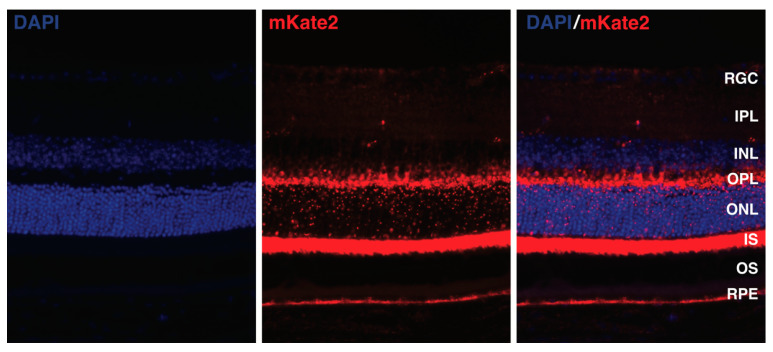
Representative section of a retina from a mito:mKate2 mouse (JAX #032188). This image displays mitochondria expressing far-red fluorescence due to the mKate2 protein, which is fused to the N-terminal of the cytochrome c oxidase subunit VIII, targeting it specifically toward mitochondria (shown in red). Nuclei are labeled with DAPI staining (blue). RGC: retinal ganglion cell; IPL: inner plexiform layer; INL: inner nuclear layer; OPL: outer plexiform layer; ONL: outer nuclear layer; IS: inner segment; OS: outer segment. RPE: retinal pigment epithelium.

**Figure 3 ijms-25-08626-f003:**
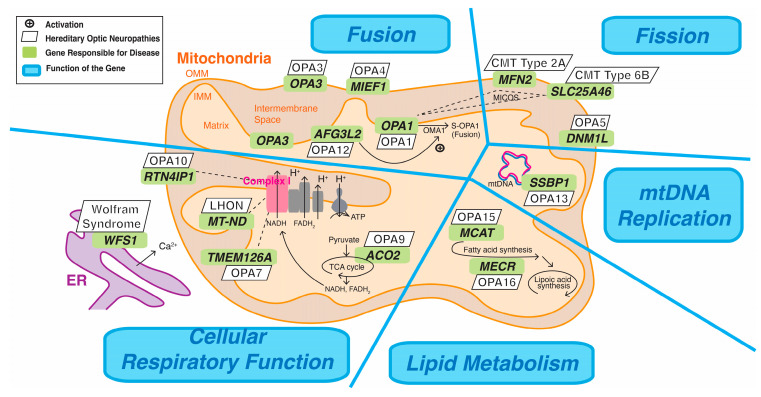
The Genetic and Molecular Pathways of Mitochondrial Dysfunction in Hereditary Optic Neuropathies. OMM, outer mitochondrial membrane. IMM, inner mitochondrial membrane; TCA, tricarboxylic acid cycle; NADH, nicotinamide adenine dinucleotide; FADH_2_, flavin adenine dinucleotide; ATP: adenosine triphosphate; OPA, optic atrophy; CMT, Charcot-Marie-Tooth Disease; LHON, Leber’s hereditary optic neuropathy; mtDNA, mitochondrial DNA; OMA1, M-AAA protease; MICOS, mitochondrial contact site and cristae organizing system.

**Figure 4 ijms-25-08626-f004:**
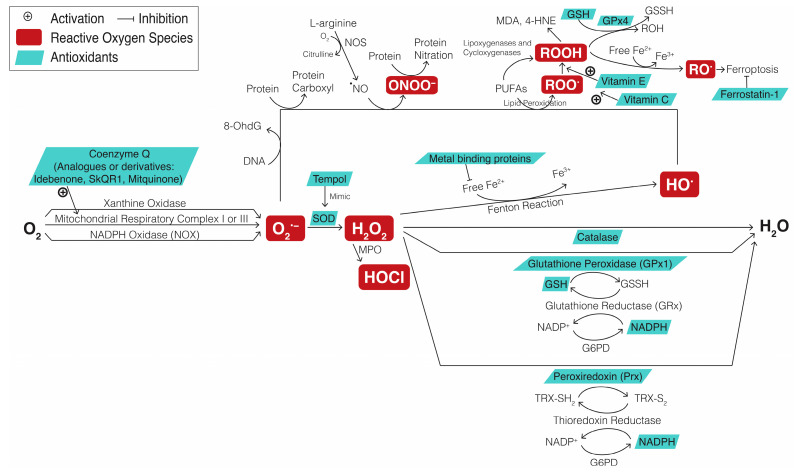
Schematic of Reactive Oxygen Species (ROS) Metabolism and Antioxidant Defense Pathway. ROS are represented by maroon text boxes, whereas antioxidants are presented in turquoise text boxes. O_2_: molecular oxygen; O_2_^•−^: superoxide anion radicals; H_2_O_2_: hydrogen peroxide; HO^•^: hydroxyl radicals; HOCl: hypochlorous acid; H_2_O: hydrogen oxide, water; ROOH: organic hydroperoxide; MDA: malondialdehyde; 4-HNE: 4-hydroxynonenal; PUFA: polyunsaturated fatty acid; NOS: nitric oxide synthase; NO^•^: nitric oxide; ONOO^−^: peroxynitrite; SOD: superoxide dismutase; G6PD: glucose 6-phosphate dehydrogenase; GSH: glutathione; GSSH: oxidized glutathione; NADPH: nicotinamide adenine dinucleotide phosphate; TRX: thioredoxin.

**Table 1 ijms-25-08626-t001:** Hereditary Optic Neuropathies: Mutated Genes and Their Encoding Physiological Functions.

Mitochondrial Function	Gene; Location; MIM	Function and Location of the Encoded Protein	Hereditary Optic Neuropathies and Their Classifications: (1) Nonsyndromic, (2) Syndromic with Optic Neuropathy as a Necessary Manifestation, (3) Syndromic with Optic Neuropathy as a Possible Manifestation, (4) Typically Does Not Manifest Optic Neuropathy
Fusion or Fission	*OPA1 (Mitochondrial dynamin-like GTPase)*; 3q29; MIM #605290	Fusion; in the IMM	(1) Optic atrophy 1 (OPA1), AD (also known as Kjer’s type optic atrophy) (MIM #165500) (2) Dominant Optic atrophy plus syndrome (DOA-plus), AD (MIM #125250) (2)Behr syndrome, AR (MIM #210000)
	*OPA3 (Outer mitochondrial membrane lipid metabolism regulator)*; 19q13.32; MIM #606580	Fusion; conflicting evidence regarding the whether OPA3 is embedded in the IMM or OMM	(2) Optic atrophy 3 (OPA3) with cataract, AD (MIM #165300) (2) 3-methylglutaconic aciduria, type III (Costeff syndrome), AR (MIM #258501)
	*AFG3L2 (AFG3-like matrix AAA peptidase, subunit 2)*; 18p11.21; MIM #604581	Fusion; in the IMM	(1) Optic atrophy 12 (OPA12), AD (MIM #618977) (4) Spastic ataxia 5, AR (MIM #614487) (4) Spinocerebellar ataxia 28 (SCA28), AD (MIM #610246)
	*MIEF1 (Mitochondrial elongation factor 1)*; 22q13.1; MIM #615497	Fusion; in the OMM	(1) Optic atrophy 14 (OPA14), AD (MIM #620550)
	*DNM1L (Dynamin 1-like)*; 12p11.21; MIM #603850	Encodes DRP1; Fission; in the OMM	(1) Optic atrophy 5 (OPA5), AD (MIM #610708) (3) Encephalopathy, lethal, due to defective mitochondrial peroxisomal fission 1, AD or AR (MIM #614388)
	*MFN2 (Mitofusin 2)*; *1p36.22*; MIM #608507	Fission; in the OMM	(3) Charcot-Marie-Tooth Disease (CMT) type 2A2A, AD (MIM #609260) (3) Charcot-Marie-Tooth Disease (CMT) type 2A2B, AR (MIM #617087) (3) Hereditary motor and sensory neuropathy, type VIA, AD (MIM #601152) (4) Lipomatosis, multiple symmetric, with or without peripheral neuropathy, AR (MIM #151800)
	*SLC25A46 (Solute carrier family 25, member 46)*; 5q22.1; MIM #610826	Cristae maintenance and fission; in the OMM	(3) Charcot-Marie-Tooth Disease (CMT) type 6B, AR (MIM #616505) (3) Pontocerebellar hypoplasia, type 1E (MIM #619303)
Cellular Respiratory Function	*MT-ND (Mitochondrial NADH dehydrogenase subunit)*; MT-ND1 3460G>A, MT-ND4 11778G>A, and MT-ND6 14484T>C accouts for approximately 90% of LHON cases.	Subunits of the NADH dehydrogenase, or known as complex I of the ETC; in the IMM	(1) Leber’s hereditary optic neuropathy (LHON) (MIM #535000)
	*TMEM126A (Transmembrane protein 126A)*; 11q14.1; MIM #612988	Assembly of complex I of the ETC; in the IMM	(2) Optic atrophy 7 (OPA7), AR (MIM #612989)
	*ACO2 (Aconitase 2)*; 22q13.2; MIM #100850	Catalyzes citrate to isocitrate in the citric acid cycle; in the mitochondrial matrix	(2) Optic atrophy 9 (OPA9), AR (MIM #616289) (4) Infantile cerebellar-retinal degeneration, AR (MIM #614559)
	*RTN4IP1 (Reticulon 4-interacting protein 1)*; 6q21; MIM #610502	Contains an oxidoreductase domain act on complex I of the ETC; in the OMM	(2) Optic atrophy 10 (OPA10) with or without ataxia, impaired intellectual development and seizures, AR (MIM #616732)
	*WFS1 (Wolframin ER transmembrane glycoprotein)*; 4p16.1; MIM #606201	Comprising mitochondria-associated endoplasmic reticulum membranes (MAM); in the endoplasmic reticulum membrane	(2) Wolfram syndrome type 1, or DIDMOAD (diabetes insipidus, diabetes mellitus, optic atrophy, and deafness), AR (MIM #222300) (2) Wolfram-like syndrome, AD (MIM #614296)
Mitochondrial DNA Replication	*SSBP1 (Single-stranded DNA-binding protein 1)*; 7q34; MIM #600439	Mitochondrial single-strand binding protein, essential for mtDNA replication; in the mitochondrial matrix	(1) Optic atrophy 13 (OPA13) with retinal and foveal abnormalities, AD (MIM #165510)
Lipid Metabolism	*MCAT (Malonyl CoA:ACP acyltransferase)*; 22q13.2; MIM #614479	Catalyze the initial step of mtFAS; in the mitochondrial matrix	(1) Optic atrophy 15 (OPA15), AR (MIM #620583)
	*MECR (Mitochondrial trans-2-enoyl-CoA reductase)*; 1p35.3; MIM #608205	Catalyze the last step of mtFAS; in the mitochondrial matrix	(1) Optic atrophy 16 (OPA16), AR (MIM #620629) (2) Dystonia, childhood-onset, with optic atrophy and basal ganglia abnormalities, AR: optic atrophy, involuntary movements, mainly dystonia (MIM #617282)

The MIM numbers for genes or phenotypes are retrieved from the Online Mendelian Inheritance in Man (OMIM) database [60]. Abbreviations: AD, autosomal dominant; AR, autosomal recessive; OMM, outer mitochondrial membrane; IMM, inner mitochondrial membrane; ETC, electron transport chain; mtFAS, mitochondrial fatty acid synthesis.

**Table 2 ijms-25-08626-t002:** Studies Investigating the Role of Oxidative Stress in RGC Degenerative Diseases.

Mechanism of Oxidative Stress Biomarker	Oxidative Stress Biomarker	Disease	Results	Ref.
Direct detection of specific ROS	Nitric oxide synthase 2	Optic neuritis	Melatonin increased nitric oxide synthase 2 level and improved RGC survival and visually evoked potential loss of optic neuritis mice	[155]
	DCFDA	Optic neuropathy	Increased H_2_O_2_ detected by DCFDA in the RGCs of an optic neuropathy animal model	[156]
	DHE	Optic nerve injury	Increased O_2_^•−^ detected by DHE in the RGCs of an optic nerve axotomy mice model	[157]
		Ocular hypertension	Increased O_2_^•−^ detected by DHE in the RGCs of an ocular hypertension animal model	[116]
Decreased antioxidants	SOD, OXPHOS complex IV protein expression	Glaucoma	Coenzyme Q10 increased SOD level and OXPHOS complex IV protein expression and promoted RGC survival in glaucomatous DBA/2J mice	[158]
	SOD, catalase, glutathione peroxidase, or glutathione reductase	Glaucoma	Decreased plasma levels of SOD, catalase, glutathione peroxidase, and glutathione reductase in patients with PAOG	[159]
	Glutathione	Glaucoma	Decreased glutathione levels in the leukocytes of patients with glaucoma	[160]
	OXPHOS complex II activity	Glaucoma	Ubiquinol, the reduced form of CoQ10, enhanced TFAM expression and OXPHOS complex II activity, as well as promoting RGC survival in glaucomatous DBA/2J mice	[161]
Upregulation of iron-binding proteins	Transferrin, ceruloplasmin, or ferritin	Glaucoma	Transferrin, ceruloplasmin, and ferritin were upregulated in a monkey model of glaucoma and human postmortem glaucomatous eyes	[162]
		Glaucoma	Higher serum ferritin levels were associated with greater odds of glaucoma	[163]
		Glaucoma	The ferroptosis inhibitor ferrostatin-1 (Fer-1) significantly enhanced RGC survival and preserved retinal function in mouse models of optic nerve crush and microbead-induced glaucoma	[164]
Cellular response to oxidative stress	4-HNE	Glaucoma	Increased lipid peroxidation indicated by 4-HNE levels in the aqueous humor of patients with glaucoma	[165]
	MDA	Glaucoma	Increased lipid peroxidation indicated by MDA levels in patients with glaucoma	[166]
		Glaucoma	Increased lipid peroxidation indicated by MDA-TBARS tests in the aqueous humor of patients with PAOG	[167]
	8-OhdG or 8-oxodG	Glaucoma	Increased oxidative DNA damage indicated by plasma 8-OHdG levels in patients with POAG	[117]
		Glaucoma	Increased oxidative DNA damage indicated by urinary 8-OHdG/creatinine associated with glaucomatous visual field progression	[168]
		Diabetic retinopathy	Nicotinamide (Vitamin B3) decreased 8-OHdG level and apoptotic RGC death in a diabetic rat model	[169]
	Protein carbonylation	Glaucoma	Increased protein oxidation indicated by protein carbonyl immunoreactivity in the retina in a chronic pressure–induced rat model of glaucoma	[170]
	*Nrf2* expression	RGC death	RGC survival in *Nrf2* knockout mice significantly lower than that in wild-type mice	[171]

Abbreviations: RGC, retinal ganglion cells; DCFDA, 2-7 dichlorofluorescein diacetate; H_2_O_2_, hydrogen peroxide; DHE, dihydroethidium; SOD, superoxide dismutase; 4-HNE, 4-hydroxynonenal; MDA, malondialdehyde; 8-OhdG or 8-oxodG, 8-hydroxy-2′-deoxyguanosine; *Nrf2*, nuclear factor erythroid 2-related factor 2; PAOG, primary angle-closure glaucoma; MDA-TBARS, malondialdehyde-thiobarbituric acid reactive substances; Ref., references.

**Table 3 ijms-25-08626-t003:** Effects of Antioxidants on Mitochondrial Dysfunction.

Antioxidant	Study Model	Results	Ref.
MitoVit E	Rat	Protected isolated rat liver mitochondria from oxidative damage	[177]
Glutathione	Rat	Delayed mitochondrial depolarization and reduced ROS generation in striatal neurons	[178]
XJB-5-131	Rat	Prevented cardiolipin oxidation in the brain and reduced neuronal death signals	[179]
CoQ10	Rat	Increased mitochondria number/density, preserved structure, and reduced ROS generation	[180]
Crocin	Human RGCs	Reduced ROS production, boosted RGC viability, and protected against apoptosis	[181]
Melatonin	Mice	Reduced loss in RGCs and caused less oxidative damage	[155]
Tempol	Mice	Limited oxidative stress and reduced neuroinflammation	[182]
CoQ10	Mice	Promoted RGC survival and blocked the apoptotic pathway in the ischemic mouse retina	[183]
SS-31 (elamipretide)	Mice	Restored redox homeostasis and improved mitochondrial quality, thereby increasing exercise tolerance	[184]
SkQR1	Rat	Reduced traumatic brain damage–related disorders of limb functions and increased survivability of neurons	[185]
MitoQ (mitoquinone)	Human NP cells	Achieved equilibrium in mitochondrial dynamics and eliminated compromised mitochondria	[186]
Ferrostatin-1 (Fer-1)	Rat	Significantly promoted RGC survival and preserved retinal function in ONC and microbead-induced glaucoma mouse models	[164]
Spearmint	Rat	Improved RGC-related ERG responses, cell density, neurotrophins, oxidative stress, and inflammation markers	[187]
Idebenone	Mice	Protected against H_2_O_2_-induced oxidative damage by reducing mitochondrial damage and autophagic activity	[188]
Trans-Resveratrol	Mice	Relieved electrophysiological injury of the retinas and inhibited RGC apoptosis	[189]
Green Tea Extract	Rat	Reduced RGC death and led to significantly higher RGC numbers and regenerated axons	[190]
Taurine	Rat	Enhanced RGC survival and protected RGCs from NMDA excitotoxicity	[191]

Abbreviations: ROS, reactive oxygen species; CoQ10, coenzyme Q10; NP cells: nucleus pulposus cells; ONC, optic nerve crush; ERG, electroretinography; H_2_O_2_, hydrogen peroxide; NMDA, N-methyl-D-aspartate; Ref., references.

**Table 4 ijms-25-08626-t004:** Effect of Promotion of Mitophagy on Mitochondrial Dysfunction.

Study Model	Intervention to Promote Mitophagy	Results	Ref.
Mice (Optic nerve injury)	Rapamycin	Increased RGC survival	[157]
Rat Neurons	Mitochondrial damage	Physiological mitochondrial damage in hippocampal axons recruits Parkin for mitophagy, a process absent in *PINK1*-mutated neurons, demonstrating the role of PINK1/Parkin pathway in local mitophagy	[197]
Mice (Obesity)	Metformin	Reduced ER stress and p53 levels to restore Parkin-mediated mitophagy in hepatocytes	[198]
Rat (Glaucoma)	Overexpression *Opa1* with AAV2-OPA1	Upregulated Parkin, leading to a greater mitochondrial surface area, better mitochondria quality, and increased fusion morphology, and increased RGC survival	[121]
Mice (Glaucoma)	Mitochondrial *Uncoupling protein 2 (Ucp2)* knock-out	Decreased ROS-mediated protein modifications and reduced RGC death	[199]
Cybrids of Human (LHON)	Rapamycin	Repaired mitochondrial defects and improved overall cell survival	[200]
Rat (NMDA-induced retinal excitotoxicity)	Metformin	Reduced RGC death and improved structure/function of RGCs	[201]
Rat (NMDA-induced retinal excitotoxicity)	Small molecule S3	Promoted RGC survival and improved mitochondrial quality	[202]
Rat (Glutamate excitotoxicity)	Fucoxanthin	Increased mitochondrial membrane potential and reduced cytotoxicity and apoptosis	[203]

Abbreviations: PINK1: PTEN-induced kinase 1; Parkin: Parkinson juvenile disease protein 2; AAV2-OPA1: adeno-associated virus 2-optic atrophy 1; LHON: Leber’s hereditary optic neuropath; Ref., references.

**Table 5 ijms-25-08626-t005:** Effect of Inhibition of Mitophagy on Mitochondrial Dysfunction.

Study Model	Intervention to Inhibit Mitophagy	Results	Ref.
*C. elegans*	RNAi against PINK1	Revealed mitochondrial biosynthetic pathway to be affected by retrograde signals through DCT-1	[204]
Mice	*Mfn2* mutation	Interruption of PINK1-Mfn2-Parkin mediated mitophagy in cardiomyocytes led to lethal cardiomyopathy and mitochondrial maturation arrest at the fetal stage	[205]
Mice	*Sesn2* mutation	Interruption of SESN2-induced mitophagy in macrophages led to increased mortality in sepsis models	[206]
Mice (Polymicrobial infection)	Lipopolysaccharide and interferon-*γ*	Interruption of PINK1-dependent mitophagy promoted microphage activation and increased cellular survival	[207]
Mice (*Opa1* mutation)	*Atg7* mutation	Normalized mitochondrial content and corrected visual loss	[208]
Mice (Human adult T-cell leukemia/lymphoma xenograft)	Chloroquine	Suppressed tumor growth and improved overall survival rate	[209]
Mice	*Pink1* mutation	Interruption of mitophagy and mitochondrial function in retinal pigment cells promoted death-resistant epithelial-mesenchymal transition	[210]

Abbreviations: DCT-1, death-associated protein kinase-related apoptosis-inducing protein kinase 1; SESN2, Sestrin2; *Mfn2*, mitofusin 2; *Opa1*, optic atrophy 1; *Atg7*, autophagy-related 7 cysteine peptidase; Ref., references.

**Table 6 ijms-25-08626-t006:** Effect of Promotion of Mitobiogenesis on Mitochondrial Dysfunction.

Study Model	Intervention	Results	Ref.
Mice	Pyruvate	Induced mitochondrial biogenesis through the pyruvate energy-sensing pathway, regulating oxidative capacity	[215]
Mice	Rosiglitazone (PPAR-γ agonist)	Improved glucose utilization, cellular function, and cognition	[216]
Human cells (Neuroblastoma)	Rosiglitazone (PPAR-γ agonist)	Increased levels of mtDNA and mitochondrial activators while maintaining constant membrane potential	[217]
Cybrids (LHON)	17β-estradiol	Increased number of mtDNA and reduced levels of ROS	[95]
Mice	AICAR (AMPK activator)	Improved biochemical phenotype and improved motor impairment	[218]
Mice	Glutathione (GSH)	Protected against brain damage and reduced loss of body weight	[219]
Human cells (Skeletal muscle of patients with type 2 diabetes and heart failure)	Epicatechin-rich cocoa	Increased the protein content and activity of mediators of mitobiogenesis and the abundance of cristae without affecting mitochondrial volume density	[220]
Human cells (Down syndrome)	Epigallocatechin-3-gallate	Promoted mitobiogenesis in Down syndrome cells and increased protein and mtDNA content	[221]
Mice	Nicotinamide riboside (NAD^+^ precursor)	Improved motor performance and exercise intolerance	[222]
Rat	Resveratrol	Promoted mitochondrial mass and DNA copy number while improving mitochondrial homeostasis and neural function	[223]
Mice and fish	miR-181a/b (microRNAs)	Protected cells from mitochondrial damage and reduced RGC death	[224]
Mice	MCT2 overexpression by viral vector	Increased RGC density and axon number, reduced energy imbalance, and increased mitochondrial function	[225]
Rat	PEDF overexpression by viral vector	Improved mitochondrial activities and induced OS-damaged cell regeneration	[226]
Mice	SIRT1 activator (NAD^+^) by electroporation gene delivery system	Delayed loss of RGC function and reduced RGC death from optic nerve injury	[227]
Mice	BX795 (Tank binding kinase 1 inhibitor)	Restored energy homeostasis and mitigated mitochondrial swelling to reduce mitochondrial damage caused by glaucoma	[212]

Abbreviations: mtDNA, mitochondrial DNA; LHON, Leber’s hereditary optic neuropathy; OS, oxidative stress; PPAR-γ, peroxisome proliferator-activated receptor gamma; AICAR, 5-aminoimidazole-4-carboxamide ribonucleotide; AMPK, AMP-activated protein kinase; MCT2, monocarboxylate transporter 2; PEDF, pigment epithelium-derived factor; SIRT1, sirtuin 1; NAD^+^, nicotinamide adenine dinucleotide, oxidized form; Ref., references.

**Table 7 ijms-25-08626-t007:** Effect of MitoPlant on Mitochondrial Dysfunction.

Study Model		Results	Ref.
In Vivo	Rat (Glaucoma)	Transplantation of bone marrow stromal cells enhanced RGC survival	[232]
In Vivo	Mice (Glaucoma)	Transplantation of human neuronal progenitor cells transfected with a vector that expresses IGF-1 increased RGC survival and increased neurite extension	[233]
In Vivo	LHON patient-derived iPSCs transplanted into immuno-deficient rats for teratoma formation assay	Cybrid technique to replace mutant mtDNA with wild-type mtDNA in iPSC-derived RGCs lowers superoxide levels and RGC apoptosis	[234]
In Vivo	Rat (Middle cerebral artery occlusion, modeling stroke)	Transplantation of MSCs induced mitochondrial repair and enhanced RGC survival	[235]
In Vivo	Rat (Optic nerve crush)	Transplantation of liver-isolated mitochondria improved respiratory capacity and increased RGC survival and regeneration capacity	[236]
In Vitro	RGC precursor-like cells	Exposure to moderate oxidative stress prior to MitoPlant enhanced mitochondrial uptake and prolonged survival of mitochondria by more than three-fold	[231]

Abbreviations: iPSC, induced pluripotent stem cell; MSC, mesenchymal stem cell; Ref., references.

**Table 8 ijms-25-08626-t008:** Clinical Trials on Pharmacological and Genetic Therapeutics in Leber’s Hereditary Optic Neuropathy (LHON) Patients.

ClinicalTrials.gov ID	Study Design	Results	Comments
NCT01793119 (RHODOS)	Randomized, double-blind. 85 patients. Intervention: idebenone (900 mg daily for 24 weeks). Control: placebo. Follow-up: 24 weeks.	Best recovery in visual acuity not statistically significant, but BCVA for patients with discordant BCVA at baseline showed significant improvement. No severe AEs reported [242]. A follow-up analysis of 70% of patients showed continued BCVA improvement even after discontinuing idebenone [243].	Demonstrated that idebenone can benefit some patients with LHON, particularly when treatment begins early. However, the modest effect highlights the need for more effective therapies.
NCT02774005 (LEROS)	Open-label, Phase IV. 199 patients with vision loss of 1–5 years since onset. Intervention: idebenone (900 mg daily for 24 weeks). Control: untreated. Follow-up: 2 years.	Clinically relevant stabilization in visual acuity was maintained in treated eyes over 2 years. Confirms the long-term efficacy of idebenone in the subacute and chronic phases. No severe AEs reported [244].	The results support the use of idebenone in slowing disease progression. However, the open-label design limits definitive conclusions.
NCT02693119	Randomized followed by OLE, phase II. Twelve patients (m.11778G>A) with decreased vision of 1–10 years since onset. Interventions: Randomized 52 weeks (1% topical elamipretide in both eyes, left, or right eye), followed OLE up to 108 weeks (both eyes). Follow-up: 4 weeks.	No significant changes in BCVA in elamipretide-treated eyes compared to vehicle eyes. No severe AEs reported [239].	Assessments of visual function beyond BCVA during the OLE and post hoc analyses merit further investigation.
NCT02652767 (RESCUE)	Open-label, Phase III. 39 patients (m.11778G>A) within 6 months of vision loss onset. Intervention: rAAV2/2-ND4, unilateral, single treatment. Control: sham injection. Follow-up: 96 weeks.	Sustained BCVA improvement was observed in both eyes over the 96-week follow-up period. No severe AEs reported [245].	Early intervention with gene therapy shows promise in halting or reversing vision loss in LHON.
NCT02652780 (REVERSE)	Open-label, Phase III. 36 patients (m.11778G>A) with 6 months to 1 year of vision loss onset. Intervention: rAAV2/2-ND4, unilateral, single treatment. Control: sham injection. Follow-up: 96 weeks.	Sustained BCVA improvement was observed in both eyes over the 96-week follow-up period. No severe AEs reported [238].	Bilateral improvement following unilateral injection of lenadogene nolparvovec was attributed to viral vector DNA transfer from the treated to the untreated eye, hypothesized to occur via the optic chiasm [238].
NCT03406104 (RESTORE)	Follow-up study for patients from REVERSE and RESCUE. Long-term (52 months). Intervention: rAAV2/2-ND4, unilateral, single treatment. Control: sham injection.	The treatment effect at last observation remained significant after adjusting for age and follow-up duration [246].	Long-term data solidify the efficacy and safety of gene therapy.
NCT03293524 (REFLECT)	Randomized, double-blind, Phase III. A total of 98 patients (m.11778G>A). Intervention: rAAV2/2-ND4, bilateral, single treatment. Control: unilateral rAAV2/2-ND4 with placebo in second eye. Follow-up: 5 years.	Bilaterally treated patients showed better improvement in BCVA and higher responder rates compared to those treated unilaterally. No severe systemic AEs; intraocular inflammation was common but mild [240,247].	Better outcomes of bilateral treatment over unilateral treatment indicate a potential dose-response effect.

Abbreviations: BCVA, best-corrected visual acuity. AE, adverse event. OLE, open-label extension.

## Data Availability

Not applicable.

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
