# Peer review of "Mitochondria in Retinal Ganglion Cells: Unraveling the Metabolic Nexus and Oxidative Stress"

_ijms, 2024, doi:10.3390/ijms25168626_

Round 1

Reviewer 1 Report

Comments and Suggestions for Authors

The review by Yang et al aims to highlight the importance of mitochondria for retinal ganglion cell function and survival. This is an important area and well worthy of an update on the literature and I was looking forward to reading it. I have to admit that I was disappointed by the amount of text unrelated to what the title promised and the omission of critical aspects that should be noted.

Overall, I believe the review could be significantly shortened as it contains many repeats. In addition, the sections about mitochondrial function (lines 62-86, 94-131, 132-158) are not related to GRC, are pretty much text book level and should be common knowledge to anyone working in biomedical or biological sciences. The same applies to the sections on oxidative stress (sections 5.2.1, 5.2.2) which should be compressed.

While Figure 2 is really helpful, the sections 5.2.1, 5.2.2 pretty much describe what can be seen in the figure. In addition, the figure is missing protein nitrosylation by the ONOO- radical, which is known to affect mitochondrial function in other systems (but not demonstrated in RGC).

For section 3.3 it is unclear how many of the papers cited are actually referring to studies in RGC cells (apart from the reference [54] that aims to pull everything together). Just because something happens in the brain does not mean it does in RGC cells. The authors rightly and repeatedly state how different these RGC are structurally and biochemically. So this generalisation is not helpful and the review should concentrate on the RGC specific evidence.

The same applies to 5.3/Table 2 (Lines 668-669): the statement that oxidative stress has an effect on RGC degenerative disease has to be approached with more caution as most of the papers listed in Table 2 describe correlations and not causation. The Himori et al paper is the single one that is most suited to support the authors statement. Therefore, I would welcome the inclusion of more papers that demonstrate causation. This should not be too difficult given that the next section deals with antioxidants to treat mitochondrial dysfunction.

BTW, I would like to encourage the authors to provide evidence that the Wolframin protein is located in the OMM as stated in line 432. The published literature and Gene cards locates WFS1 into the ER membrane and not the OMM (https://www.genecards.org/cgi-bin/carddisp.pl?gene=WFS1#localization ) and antibody stains support this localisation.

In contrast to generalised sections unrelated to the specific topic, the review is missing important information:

1.       Table 4: This table is missing crucial information. HOW was mitophagy promoted in each study? I would rather remove the author names from the table (this applies to all tables) as the reference numbers are sufficient.

2.       Table 5 same comment as above

3.       Section 6.3: the statement “A variety of compounds can stimulate mitobiogenesis” suggests that in the corresponding table the reader learns which ones are referred to. Unfortunately, again this table 6 does not provide this information

4.       Table 7: The use of MitoPlant is exciting but it is important to highlight, which of the studies have actually been using GRC (and critically concentrate on those). The table should state if the experiments were done in vitro or in vivo to get a sense if this technology is translatable and list studies that were negative or showed adverse effects.

5.       I am surprised that in the section on glaucoma the mitochondrial angle was not elaborated on more. There is for example evidence of systemic mitochondrial complex 1 dysfunction in glaucoma patients (https://pubmed.ncbi.nlm.nih.gov/22427588/ ) or reduced expression of OPA1 in POAG, or the higher than expected frequency of OPA1 polymorphisms in POAG and especially in NTG patients. As such the review does not provide any in depth assessment of the interactions. This is also illustrated by the next point.

1.       If mitochondrial function is so important and especially complex 1 function (evidenced by LHON and glaucoma) and if the structure (non-myelinated versus myelinated) of the GRCs is one major aspect of the mitochondrial dependence of RGC, how can we explain that in LHON we see central vision loss (with the small bundles affected predominantly) while in glaucoma we see mostly peripheral vision loss? I would have expected that the review not only deals with the supporting evidence but also highlights the problems and lack of knowledge with this association.

2.       In this context the review would benefit from a graphical representation of RGC that include the eye, the optic nerve, regions of myelination, and mitochondrial density in the different regions to highlight the authors point that the structure of the RGC is unique and contributes to their vulnerability.

3.       Finally, a major omission in this review is the lack of cited clinical data to support the title and content of the review. For example, several clinical trials have been performed in LHON patients with gene therapy and idebenone. Idebenone is marketed in Europe for LHON which suggests that it has some activity (admittedly not very much). Clinical trials have also been performed for some of the other small molecules listed in this review but are completely ignored as supportive evidence (or not) for the main topic of this review. For example, did the clinical trials work or not? What can we learn from the results and how do the results reflect back on the thesis of this review that mitochondrial are central to RGC function and health?

Overall, removing text book information from the review and replacing it with studies relevant to the review title and a much more detailed and in depth, critical assessment with pros and cons for interventions would increase the value of this review.

Reviewer 2 Report

Comments and Suggestions for Authors

The manuscript "Mitochondria in Retinal Ganglion Cells (RGC): Unravelling the Metabolic Nexus and Oxidative Stress" is a well-written and informative review that covers essential aspects of mitochondrial biology in RGCs. My concerns and comments below:

1.    Certain sections would benefit from a deeper analysis. For example, the discussion on mitochondrial dynamics (fusion and fission) could be expanded to provide a clearer understanding of how these processes influence RGC function and survival.

2.    The manuscript would be enhanced by the inclusion of additional figures or diagrams that illustrate key concepts, such as the mitochondrial electron transport chain, dynamics of mitochondrial fusion and fission, and pathways of oxidative stress.

3.    While the manuscript mentions novel therapeutic approaches, it would benefit from a more detailed discussion on the current challenges, limitations, and future directions of these therapies. This would provide a more balanced and critical perspective.

4.    The integration of recent studies and reviews into the manuscript could be improved. Providing a critical comparison of different research findings would add depth and context to the discussion.

5.     The manuscript could be more effectively organised with clearer subheadings and transitions between sections. This would improve the readability and flow of information, making it easier for readers to follow the complex topics discussed.

6.    A detailed exploration of mitochondrial fusion and fission processes and their regulatory mechanisms in RGCs is needed. Please discuss how disruptions in these processes contribute to disease pathogenesis, including specific molecular pathways and proteins involved.

7.    Please provide a thorough analysis of the limitations and challenges associated with mitochondrial-based therapies and gene therapies. Kindly discuss factors such as delivery mechanisms, specificity, potential off-target effects, and the current status of clinical trials.

8.    Please critically analyse and incorporate findings from recent studies. Compare and contrast different therapeutic strategies, highlighting their efficacy, potential, and shortcomings. This will provide a comprehensive overview of the current research landscape.

9.    Please reorganise the manuscript to create a more logical and coherent structure and use distinct subheadings for different topics and ensure smooth transitions between sections to maintain a cohesive narrative.

Comments on the Quality of English Language

Minor syntax and grammatical errors. 

Reviewer 3 Report

Comments and Suggestions for Authors

The manuscript “Mitochondria in Retinal Ganglion Cells: Unraveling the Metabolic Nexus and Oxidative Stress” by Yang is a review article which explored the role of mitochondria in retinal ganglion cells (RGCs), which are thought to be essential for visual processing. This review highlighted the importance of mitochondria in providing metabolic support, regulating cellular health, and responding to cellular stress in RGCs. Because promising research initiatives and novel technologies are emerging, including mitochondrial-based therapies, gene therapies, and mitochondrial transplantation, these advances can offer potential strategies for addressing mitochondrial dysfunction in the retina. In general, this review article is critical in this field and contains essential contents. However, I have several comments before this manuscript is accepted for publication.

1. The abstract is too short. Please add the description about metabolic nexus and oxidative stress.

2. Figure 2: The explanation of Figure 2 is described in the text, but it is difficult for the readers to focus on which part of figure 2.
